# PEMs: Pre-trained Epidemic Time-Series Models

## Abstract

Providing accurate and reliable predictions about the future of an epidemic is an important problem for enabling informed public health decisions. Recent works have shown that leveraging data-driven solutions that utilize advances in deep learning methods to learn from past data of an epidemic often outperform traditional mechanistic models. However, in many cases, the past data is sparse and may not sufficiently capture the underlying dynamics. While there exists a large amount of data from past epidemics, leveraging prior knowledge from time-series data of other diseases is a non-trivial challenge.

Motivated by the success of pre-trained models in language and vision tasks, we tackle the problem of pre-training epidemic time-series models to learn from multiple datasets from different diseases and epidemics. We introduce Pre-trained Epidemic Time-Series Models (PEMs) that learn from diverse time-series datasets of a variety of diseases by formulating pre-training as a set of self-supervised learning (SSL) tasks. We tackle various important challenges specific to pre-training for epidemic time-series such as dealing with heterogeneous dynamics and efficiently capturing useful patterns from multiple epidemic datasets by carefully designing the SSL tasks to learn important priors about the epidemic dynamics that can be leveraged for fine-tuning to multiple downstream tasks. The resultant PEM outperforms previous state-of-the-art methods in various downstream time-series tasks across datasets of varying seasonal patterns, geography, and mechanism of contagion including the novel Covid-19 pandemic unseen in pre-trained data with better efficiency using smaller fraction of datasets.

## 1 Introduction

Predicting the trends of an ongoing epidemic is an important public health problem that influences real-time decision-making affecting millions of people. Forecasting of time series of important epidemic indicators is a well-studied challenging problem (Rodríguez et al., 2022b; Chakraborty et al., 2018). Availability of traditional as well as novel datasets such as testing records, social media, etc. that capture multiple facets of the epidemic as well as advances in machine learning and deep learning in particular have enabled to build models that learn from these datasets and show promising results, often outperforming traditional mechanistic methods (Cramer et al., 2021; Reich et al., 2019).

Many public health and research initiatives collect data from various diseases over many decades at various spatial granularities in different geographies. Initiatives such as Project Tycho (van Panhuis et al., 2018) have aggregated these datasets that date back to the 1880s. However, when building a data-driven model for an ongoing epidemic, we typically train the model *only* using the past data of the epidemic without leveraging useful knowledge information and patterns from past epidemics. In contrast, learning pipelines for language and vision use pre-trained models trained on a much larger dataset (Qiu et al., 2020; Du et al., 2022; Gunasekar et al., 2023). These pre-trained models are fine-tuned to the task of interest. The benefits of the pre-trained models are two-fold. First, the pre-trained weights are good initialization for faster and more effective training. Moreover, pre-trained models learn useful underlying structures and patterns from larger pre-trained datasets such as common syntactic and semantic knowledge in the case of language and the ability to recognize useful patterns in the case of vision. Initiating training from these pre-trained models usually results in faster training and better performance.

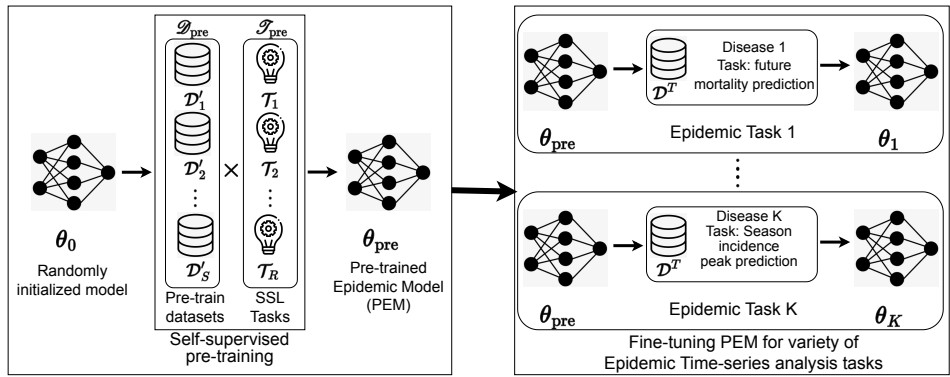

Figure 1: SSL based pre-training of PEM over multiple tasks $\mathcal{T}_{\mathrm{pre}}$ and multiple pre-train datasets $\mathcal{D}_{\mathrm{pre}}$

Inspired by the success of pre-training in these fields, our goal is to leverage a large quantity of past epidemic time-series datasets and build general pre-trained models that can be fine-tuned for multiple epidemic forecasting tasks on new unseen diseases. However, there are challenges specific to time-series data that make pre-training non-trivial. First, unlike text or images, epidemic time-series data is very heterogeneous. Each dataset records different kinds of epidemic targets such as cases, mortality, hospitalization rates, etc., produced by different underlying dynamics from different geographical origins and can be affected by various levels of noise. Moreover, while the size of the dataset potentially available for pre-training is typically larger than training data, it is still orders of magnitude smaller than in the case of text and images. Indeed, previous state-of-art epidemic forecasting models have leveraged various methods such as sequence similarity Adhikari et al. (2019); Wang et al. (2020), geographical relations Deng et al. (2020); Wu et al. (2018) using mechanistic priors Rodríguez et al. (2022a); Gao et al. (2021); Wang et al. (2021) ensembling Cramer et al. (2021); McAndrew & Reich (2021), etc. to overcome challenges related to data sparsity and dealing with noise. However, they can not effectively leverage heterogeneous datasets from other diseases and only use training data specific to the epidemic. Moreover, we do not have access to reliable auxiliary data such as mechanistic priors or spatial relations in most real-world epidemics. Therefore, the pre-training methods we design for epidemic time-series datasets should be data-efficient and effective in learning useful general information about the heterogeneous epidemic datasets. We tackle these challenges by viewing the problem of pre-training on multiple disease datasets via a self-supervised learning (SSL) framework.

While there have been multiple recent works on SSL for time-series data (Yue et al., 2022; Tonekaboni et al., 2021; Eldele et al., 2021; Franceschi et al., 2019), these methods cannot be adapted to pre-training on multiple diseases. Instead, they perform SSL on the training dataset specific to the task. Therefore, they cannot leverage useful epidemic priors during pre-training from multiple disease datasets. Our work, in contrast, focuses on providing general-purpose pre-trained epidemic models that can be fine-tuned to a wide range of epidemic analysis tasks for different diseases including novel and unseen epidemics by pre-training on datasets from various diseases of different periodicity, sparsity, and the nature of underlying epidemic dynamics from the past. We design three SSL tasks to leverage such multi-disease time-series datasets and learn pre-trained models called PEMs (Pre-trained Epidemic Time-Series Models). We also design our transformer-based architecture to enable us to better capture the temporal structure of time-series and deal with datasets of varying magnitude and data distributions. We show that PEMs can be fine-tuned to perform a wide variety of forecasting tasks for multiple diseases including novel diseases such as Covid-19 that are unseen in pre-training. Our contributions are summarized as follows: **(A) General pre-trained model for epidemic analysis tasks** We are the first to propose a single pre-trained model trained on multiple disease datasets that can be fine-tuned to a wide variety of downstream epidemic tasks concerning wide range of diseases. **(B) Self-supervised pre-training on cross-disease datasets**: We propose carefully designed SSL tasks that can learn from pre-train datasets of multiple diseases and efficiently capture important patterns of epidemic dynamics such as seasonality, and behavior around significant periods. We also note that, unlike text data, each time-stamp may not have sufficient semantic information. Similar to Nie et al. (2022), we propose feeding segments of the time series as individual tokens to better

capture local temporal trends. **(C) State-of-the-art forecasting and peak prediction performance on multiple datasets**: We evaluate PEM on disease datasets of different characteristics such as seasonality, geography, and mode of infection. We observed an 11-24% improvement in performance over previous state-of-the-art baselines and SSL methods. PEM also outperforms other methods on the task of forecasting mortality during the novel Covid-19 pandemic. **(D) Significant improvement in data and training efficiency and adaptability to novel epidemics**: PEM also outperforms the baselines after significantly smaller training time and using smaller training data size compared to baselines. We also perform detailed ablation showing the importance of each of the proposed SSL tasks and modeling choices.

## 2 PRELIMINARIES

**Epidemic analysis tasks**   We focus on multiple tasks associated with forecasting on time-series of epidemic indicators of various diseases. Informally, given the past time-series of an indicator of the epidemic such as case counts or mortality, the goal is to predict specific characteristics that inform the future dynamics of the epidemic. These targets include future values of the indicators (forecasting), predicting the time and magnitude of the peak or onset of the epidemic.

Formally, let $\mathcal{D}^T$ be time-series data of epidemic indicators. Let the time-series from time-stamp 1 to $T$ be denoted as $\mathbf{y}^{(1\dots T)} \in \mathbb{R}^T$. The goal of an epidemic analysis task is to predict some useful property of the future values of the time-series $\mathbf{y}^{(T+1,\dots,T+K)}$. For example, the task of forecasting involves predicting the values of $\mathbf{y}^{(T+1,\dots,T+K)}$. Similarly, peak time prediction involves the prediction of time $t' = \arg\max_{t \in T+1\dots T+K} y^{(t)}$ and peak intensity prediction involves estimating the value at the peak $\max_{t \in T+1\dots T+K} y^{(t)}$.

**Self-supervised pre-training for Epidemic analysis**   All previous methods only use datasets relevant to the specific epidemic to train or calibrate their models. However, modelers typically also have access to time-series from other diseases collected in the past and aggregated by initiatives like Project Tycho van Panhuis et al. (2018). Our work focuses on leveraging useful characteristics of epidemic data from these cross-disease data sources to provide general pre-trained models that can better adapt to various epidemic analysis tasks.

Formally, along with the dataset $\mathcal{D}^t$ relevant to the specific epidemic of interest, we also have access to time-series datasets of other diseases from the past. Let these set of datasets be denoted by $\mathcal{D}_{\text{pre}} = \{\mathcal{D}'_1, \mathcal{D}'_2, \dots, \mathcal{D}'_S\}$. We call $\mathcal{D}_{\text{pre}}$ as *pre-train datasets*. Each of $\mathcal{D}'_j$ are univariate time-series datasets. We use the Self-supervised learning (SSL) framework to leverage $\mathcal{D}_{\text{pre}}$ for pre-training. We introduce a set of tasks $\mathcal{T}_{\text{pre}} = \{\mathcal{T}_i\}_{i=1}^R$. At a high level, each task takes a time-series from $\mathcal{D}_{\text{pre}}$, transforms it, and trains the model to retrieve important properties of the input time-series. These properties include identifying seasons of certain segments of time-series, reconstructing important parts of the time-series, etc.

**Problem Statement**   *Given heterogeneous pre-train datasets $\mathcal{D}_{pre}$ from multiple diseases, we aim to learn useful patterns, epidemic dynamics, and knowledge from $\mathcal{D}_{pre}$ via SSL tasks $\mathcal{T}_{pre}$ such that the resultant pre-trained model can be fine-tuned to provide better performance on any epidemic analysis tasks for unknown diseases leveraging the generalizable patterns learned from $\mathcal{D}_{pre}$.*

## 3 METHODOLOGY

**Overview**   Our pipeline of leveraging pre-trained models for downstream epidemic tasks resembles that used in NLP and Vision problems. Let $M(\theta_{pre})$ be the base PEM used for various epidemic analysis tasks and parameterized by $\theta_{pre}$. $M$'s parameters are simultaneously pre-trained on each of the SSL tasks in $\mathcal{T}_{\text{pre}}$ on all the pre-train datasets. This allows $M$ to learn from the underlying epidemic dynamics of multiple diseases without explicit supervision. We then fine-tune the pre-trained $M(\theta_{pre})$ by first appending appropriate output layer $G(\theta_{last})$ to the model based on the task and train all the $M$ and $G$ for the given downstream task.

### 3.1 SEGMENTED TRANSFORMER MODEL

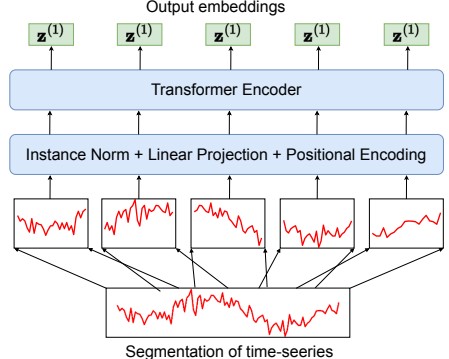

Output embeddings

Transformer Encoder

Instance Norm + Linear Projection + Positional Encoding

Segmentation of time-seeries

Figure 2: Segmented Transformer Architecture

Transformers (Vaswani et al., 2017) have been widely used in modeling sequential data, especially text data, and form the backbone architecture of large pre-traned models. This is due to their ability to model long-range temporal relations as well as scale up to learn from large datasets during pre-training (Dosovitskiy et al., 2020; Brown et al., 2020). Recent works (Zhou et al., 2021; Chen et al., 2021) have shown the efficacy of transformers for time-series forecasting in a wide range of domains. Therefore, we design a transformer-based architecture for PEM s and modify it to better capture local temporal patterns. However, our framework can be easily extended to other neural sequential models like RNNs and convolutional networks.

Most previous works input features from each time step as a single token. However, unlike text data, each individual time-stamp may not provide enough semantic meaning about temporal patterns of the time-series. Therefore, similar to Nie et al. (2022), we use segments of time-series, instead of individual time-stamps as input tokens. We first segment the input time series $\mathbf{x}^{(1:T)} \in \mathbb{R}^T$ into uniform segments of size $P$ with stride length $S$ resulting in a sequence of length $L = \lfloor \frac{T-P}{S} \rfloor + 1$ denoted by $\hat{\mathbf{x}}^{(1:L)} \in \mathbb{R}^{P \times L}$. Along with better temporal modeling, segmenting the input can also enable the transformer model to efficiently process sequences of longer lengths since its inference speed scales quadratically with sequence length.

For the first layer of the model, we project all the features of all the time stamps of each segment into an embedding and also inject positional information for each segment. We first pass each segment $\hat{\mathbf{x}}^{(l)}$ of the sequence $\hat{\mathbf{x}}^{(1:L)}$ through a linear layer and add positional embedding to its output:

$$\mathbf{u}^{(l)} = \mathbf{W}_1 \hat{\mathbf{x}}^{(l)} + \text{pos}(l) \tag{1}$$

where $\mathbf{W}_i \in \mathbb{R}^{P \times D}$ and $\text{pos}(\cdot)$ is the positional encoding defined as

$$\text{pos}(l) = \begin{cases} \sin(l/10^{5l/D}) & \text{if } l \text{ is even} \\ \cos(l/10^{5(l-1)/D}) & \text{if } l \text{ is odd.} \end{cases} \tag{2}$$

Positional encodings, therefore, help the model identify the absolute positions of the input segments. The encodings $\{\mathbf{u}^{(l)}\}_{l=1}^{L}$ are fed into stacks of multi-head attention layers of the transformer similar to Vaswani et al. (2017) and we finally receive the output as a sequence of embeddings $\mathbf{z}^{(1:L)} \in \mathbb{R}^{D \times L}$. For a specific SSL task or downstream fine-tuning tasks, we can append appropriate layers on top of $\mathbf{z}^{(1:L)}$ to generate outputs of desirable dimensions and properties.

## 3.2 SELF-SUPERVISED LEARNING

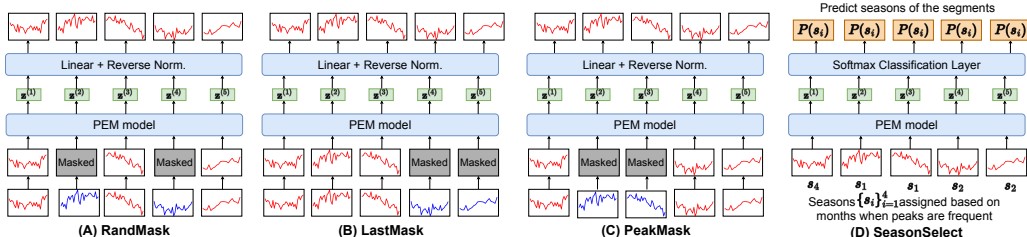

Figure 3: SSL task of $\mathcal{T}_{\text{pre}}$ are designed to efficiently capture important characteristics of epidemic dynamics by training on multiple disease datasets $\mathcal{D}_{\text{pre}}$.

The goal of pre-training is to provide the model with useful latent information about epidemic dynamics from multiple disease datasets. Unlike the case of text or images, we do not have a large pre-train dataset. Moreover, the datasets for each disease is very heterogenous with each disease

time-series measuring different epidemic indicator relevant to the disease and having large temporal and spatial variance. Therefore, we require our SSL tasks to be designed such that useful information from each of the disease dynamics is effectively captured. We present the following four SSL tasks that capture various aspects of epidemic dynamics.

**RANDMASK: Random Masking**   Epidemic forecasting models usually need to interpolate or extrapolate from input time-series to infer important characteristics of the epidemic. Similar tasks have been proposed on individual text tokens for language models (Devlin et al., 2018) to learn from large quantity of unlabeled text data. Random masking has also been explored for representation learning in previous works (Zerveas et al., 2021; Nie et al., 2022). However, they are applied on the same dataset as that used for training unlike our data and task-agnostic pre-training setup. For our application, random masking allows the model to learn important patterns observed in typical epidemic time series. Formally, given the segment inputs $\hat{\mathbf{x}}^{(1:L)}$, we randomly mask $\gamma$ fraction of the segments with zero values and input to the model. We apply a two-layer feed-forward network on top of each $\mathbf{z}^{(l)}$ to predict back all the values of segments for both masked and unmasked segments. We use mean-squared error (MSE) loss to train for this task which we denote as RANDMASK ($\gamma$).

**LASTMASK: Last segments masking**   Since most epidemic tasks involve predicting properties of the future of the epidemic including forecasting future values of the input time-series, we propose another SSL task LASTMASK that aims to re-construct the most recent segemnts of the input time-series. Formally, LASTMASK ($\gamma$) is similar to RANDMASK ($\gamma$), except instead of masking random $\gamma$ fraction of the segments, we mask only the last $\gamma$ fraction of the segments. LASTMASK, therefore, helps enable the model to predict the future dynamics of the epidemic by observing the past.

**PEAKMASK: Masking around the peak**   Due to the chaotic nature of the epidemic curve around the peak, prediction of the shape of the curve around the peak is particularly hard. Epidemic curves can peak due to many expected as well as unforeseen situations such as seasonal shifts due to weather, sudden outbreak, the introduction of new variants of the pathogen strain or shifts in human behavior and policy changes. PEM needs to learn to anticipate and adapt to such scenarios. Therefore, we introduce the PEAKMASK task which is another variant of RANDMASK. For each input time series, we identify the time stamp with the maximum value. We then mask all the segments that cover the maximum value's time-stamp and train the model to recover these segments.

**SEASONDETECT: Seasonal detection**   Many diseases like influenza, chickenpox, Lyme disease, etc have seasonal patterns and peak at specific seasons. Identifying the seasonal shifts in disease dynamics is important and valuable for many epidemic analysis tasks such as forecasting and peak detection. We, therefore introduce SEASONDETECT, an SSL task for detecting the season of each time-series segment. We first divide the 12 months of the year into 4 seasons: Season 1 (Dec-Feb), Season 2 (Mar-May), Season 3 (June-Aug), and Season 4 (Sept-Nov). For each of the seasonal diseases $d$, in the pre-train dataset, we detect the season with the most peaks by calculating the month with the highest value in the time series for all years in the dataset. We call this the peak season $s_1(d)$. We then label the rest of the diseases relative to the peak season: $s_2(d)$ comes after $s_1(d)$, $s_3(d)$ comes after $s_2(d)$ and finally $s_4(d)$ comes after $s_3(d)$. For example, in case of $d =$ Influenza in US, $s_1(d)$ is assigned to months in Sept-Nov, $s_2(d)$ in Dec-Feb and so on.

The goal of the SEASONDETECT task is to identify the correct season among $\{s_i(d)\}_{i=1}^4$ for each of the input segments. If a segment contains months from two seasons, it is assigned to the season which has the majority of the time steps (in case of a tie, the chronologically first season is assigned). This problem is a segment classification problem. Therefore, we apply a classification layer on top of each output embedding $\mathbf{z}^{(l)}$ that outputs the logits for the four seasons. We use cross-entropy as the loss function for this task.

### 3.3   OTHER PRE-TRAINING AND FINE-TUNING DETAILS

**Dataset and Instance normalization**   The values of the time series of each dataset can vary widely. For example, the ILI (influenza-related illness) indicators released by CDC are in the range of 0-10 whereas, for other disease datasets such as Covid-19 or typhoid raw numbers on mortality or hospitalization are reported. Therefore, as part of pre-processing we first normalize the time-series of each dataset of pre-train datasets independently.

Moreover, the data distribution and the magnitude of the time-series can vary across time for the same disease. Therefore, we use reversible instance normalization (Kim et al., 2021) that performs instance normalization on the input time series and reverses the normalization of the output of the model.

**Multi-task training of all SSL tasks**   We pre-train our PEM with all four SSL tasks on all pre-train datasets $\mathcal{D}_{pre}$ together. For each of the SSL tasks, we have a separate final layer but the parameters of the PEM are shared across all tasks. This is similar to hard parameter sharing used in multi-task learning (Caruana, 1998). For each batch, we randomly choose a dataset from $\mathcal{D}_{pre}$ and randomly sample time-series from the dataset. If the chosen dataset's disease is not seasonal, we skip training for SEASONDETECT for that batch. This allows PEM to learn from all the tasks, each of which imparts useful information from $\mathcal{D}_{pre}$ which have varying utility for different downstream tasks.

**Linear-probing and fine-tuning**   Kumar et al. (2022) showed that fine-tuning all the parameters of the pre-trained model for a specific downstream task can perform worse than just fine-tuning only the last layer (linear probing), especially for downstream tasks which are out-of-distribution to pre-trained data. Since the downstream tasks in our case involve fine-tuning on novel diseases, PEM suffers from this effect as well. Therefore, based on their recommendation, we perform a two-stage fine-tuning process: we first perform linear probing followed by fine-tuning all the parameters.

## 4 EXPERIMENTS

**Datasets**   We leverage a large number of epidemic datasets aggregated by Project Tycho (van Panhuis et al., 2018). It has datasets from 1888 to 2021 for different diseases collected at state and city levels in the US. While most of the datasets are collected on a weekly basis, many of these are very sparse and have missing data. We collect all the datasets that have time-series of length at least 10 to remove sparse data with short time-series. We also use the weekly influenza data for US and Japan collected by CDC and NIID respectively. Specifically, we use the aggregated and normalized counts of outpatients exhibiting influenza-like symptoms released weekly by CDC[1]. For influenza in Japan, we use influenza-affected patient counts collected by NIID[2]. In total, we have 11 diseases: Hepatitis A, measles, mumps, pertussis, polio, rubella, smallpox, diphtheria, influenza, typhoid and Cryptosporidiosis (Crypto.). We used all the data in Project Tycho till the year 1980 for each of these diseases for pre-training. Since the influenza datasets are more recent and are collected from 2001 and 2010 for US and Japan respectively, we use influenza data up to 2012 from both countries for pre-training. This set of disease datasets captures a rich variety of epidemic dynamics such as seasonality, mode of spread, underlying biology, etc.

**Real-time Epidemic analysis tasks**   We evaluate the PEM's performance on multiple diseases. First, we evaluate forecasting for weekly influenza incidence in US and Japan, two geographically distinct locations, from 2013 to 2020. We also perform forecasting on Cryptosporidiosia (Crypto.) from 2006 to 2012 and from 1980 to 1985 for typhoid. Note that these are very diverse diseases with very different inherent dynamics: influenza is air-borne, and cryptosporidiosis and typhoid are water-borne. While influenza and cryptosporidiosis are seasonal (Flu peaks during the winter while cryptosporidiosis peaks during the summer), typhoid has a stable and low incidence except for a few sudden outbreaks (Matsubara et al., 2014). We forecast the disease indicators at the national level for the entire country (US or Japan) for up to 4 weeks into the future. Note that we use a real-time forecasting setup (Reich et al., 2019; Adhikari et al., 2019) for training the model: for forecasting up to four weeks from the current week, we use all the data till the current week to fine-tune the PEM. We also perform peak week and intensity prediction for influenza similar to previous epidemic initiatives (Reich et al., 2019). For each week in the season, we train the model to predict the week at which the peak occurs and the magnitude of the peak. Note that the epidemic curve may have already peaked in the past when we train the model at the current week but the ground truth can only be known after the full season. Since the ground truth is a real number or an integer, we use the root mean squared error (RMSE) to evaluate all the tasks similar to (Adhikari et al., 2019; Kamarthi et al., 2021). (note that cross-entropy loss is used for peak week prediction due to discrete ground truth).

---

[1]`https://gis.cdc.gov/grasp/fluview/fluportaldashboard.html`
[2]`https://www.niid.go.jp/niid/en/idwr-e.html`

**Baselines** We compare PEM with recent state-of-the-art (SOTA) models for both general time-series forecasting as well as models that are designed for epidemic analysis tasks. The six recent state-of-the-art forecasting baselines are: • INFORMER (IF) (Zhou et al., 2021) : Proposes an efficient sparse self-attention mechanism and a distillation mechanism to focus on the most important time-stamps. • AUTOFORMER (AF) (Chen et al., 2021): Replaces self-attention with an auto-correlation mechanism to efficiently capture temporal dependencies. • PATCHTST (PT) (Nie et al., 2022): The backbone architecture used by PEM that uniformly segments the input time-series into individual tokens of the transformer. Note that we do not perform any pre-training for PT baseline. • DLINEAR (DL) (Zeng et al., 2023): Uses linear layers instead of transformers for forecasting with comparable performance. • TIMESNET (TN) (Wu et al., 2023): Uses a novel 2D inception block to model multiple temporal variations and periodicity in time-series. • MICN (Wang et al., 2022): Proposes multiple convolutional layers that capture patterns at multiple scales and merge them. We also compare against the following previous SOTA time-series epidemic forecasting models: • EPIDEEP (ED) (Adhikari et al., 2019): Leverages similarity between current and historical time-series to provide interpretable forecasts • EPIFNP (EF) (Kamarthi et al., 2021): A previous state-of-art model for calibrated accurate forecasting extending Neural process framework Louizos et al. (2019) for sequential data. • FUNNEL (FL) (Matsubara et al., 2014) : Flexible Mechanistic model that can capture and forecast multiple epidemics by modeling useful characteristics like seasonality and sudden outbreaks • EBAYES (EB) (Brooks et al., 2015): Used Empirical Bayes framework for flue forecasting and won previous FluSight competitions Reich et al. (2019).

## 5 RESULTS

We evaluate PEM through the following questions: **Q1:** Does PEM provide state-of-art performance in various epidemic analysis tasks? **Q2:** Does pre-training enable faster and efficient training? **Q3:** Is PEM efficient in using less training data to provide consistently superior performance? **Q4:** How does PEM perform on diseases not available during pre-training? **Q5:** How does PEM compare against previous self-supervised time-series forecasting methods? **Q6:** How do each of the SSL tasks and various other modeling choices affect the performance of PEM? We provide additional details on hyperparameters and training details in the Appendix §B. and a link to code and datasets[3].

Table 1: Weekly forecasting performance (RMSE) of PEM and other top general forecasting and epidemic forecasting baselines. Top scores are in **bold** and the second best are underlined. We observe the evaluation metrics scores to be statistically significantly better for PEM using pair-wise t-test ($p \leq 0.05$) over 10 random runs.

| Datasets | Week ahead | General-time Series | | | | | | Epi-Specific | | | | PEM |
|---|---|---|---|---|---|---|---|---|---|---|---|---|
| | | AF | IF | PT | DL | TN | MICN | EF | ED | EB | FUNNEL | |
| **Flu-US** | 1 | 1.17 | 1.23 | 0.47 | 0.56 | 0.48 | 0.46 | 0.42 | 0.68 | 1.18 | 1.31 | **0.39** |
| | 2 | 1.28 | 1.37 | 0.83 | 0.72 | 0.51 | 0.52 | 0.48 | 0.73 | 1.26 | 1.33 | **0.42** |
| | 3 | 1.55 | 1.74 | 0.94 | 1.19 | 0.74 | 0.65 | 0.79 | 1.14 | 1.27 | 1.34 | **0.58** |
| | 4 | 1.64 | 2.11 | 1.16 | 1.25 | 0.97 | 0.81 | 0.78 | 1.81 | 1.34 | 1.37 | **0.61** |
| | Avg | 1.41 | 1.61 | 0.85 | 0.93 | 0.68 | 0.61 | 0.62 | 1.09 | 1.26 | 1.34 | **0.50** |
| **Flu-Japan** | 1 | 1139 | 1227 | 1205 | 944 | 922 | 934 | 992 | 1186 | 1172 | 1388 | **831** |
| | 2 | 1572 | 1503 | 1517 | 1147 | 951 | 948 | 1058 | 1395 | 1486 | 1694 | **894** |
| | 3 | 1676 | 1814 | 1667 | 1359 | 1189 | 1074 | 1179 | 1573 | 1858 | 1934 | **1035** |
| | 4 | 2044 | 1857 | 1918 | 1538 | 1488 | 1422 | 1572 | 1634 | 2297 | 2145 | **1069** |
| | Avg | 1608 | 1600 | 1577 | 1247 | 1138 | 1095 | 1200 | 1447 | 1703 | 1790 | **957.25** |
| **Crypto.** | 1 | 177 | 166 | 181 | 193 | 211 | 227 | 176 | 211 | 205 | 229 | **147** |
| | 2 | 194 | 197 | 257 | 238 | 236 | 267 | 195 | 259 | 381 | 415 | **176** |
| | 3 | 246 | 294 | 306 | 276 | 289 | 311 | 224 | 327 | 496 | 614 | **205** |
| | 4 | 314 | 349 | 395 | 328 | 341 | 369 | 259 | 411 | 642 | 665 | **239** |
| | Avg | 233 | 252 | 285 | 259 | 269 | 294 | 214 | 302 | 431 | 481 | **192** |
| **Typhoid** | 1 | 3.25 | **2.97** | 3.73 | 3.35 | 3.11 | 3.08 | 3.65 | 4.84 | 4.39 | 4.77 | 3.02 |
| | 2 | 4.19 | 3.93 | 5.78 | 4.06 | 3.76 | 3.66 | 3.97 | 5.11 | 5.33 | 5.13 | **3.38** |
| | 3 | 6.44 | 4.66 | 5.94 | 4.44 | 4.51 | 4.27 | 5.12 | 7.36 | 8.87 | 9.28 | **4.02** |
| | 4 | 6.98 | 5.19 | 6.94 | 5.13 | 5.22 | 4.42 | 5.93 | 8.95 | 11.98 | 13.22 | **4.61** |
| | Avg | 5.22 | 4.19 | 5.60 | 4.25 | 4.15 | 3.86 | 4.67 | 6.57 | 7.64 | 8.10 | **3.76** |

**Forecasting performance (Q1)** We perform real-time forecasting on four diseases discussed in §4 for 1-4 weeks ahead and observe the performance in Table 1. We observe that PEM's average

---

[3]Anonymized code link: `https://anonymous.4open.science/r/EmbedTS-3F5D/`

forecasting performance outperforms both general time-series forecasting baselines as well as models specifically designed for epidemic forecasting. On average, PEM provides 11-24% improvement in RMSE. The performance improvements are larger with longer forecast horizons.

We also perform prediction of time and intensity of onset and peak of the epidemic curves of seasonal diseases such as influenza as done in past Flusight forecasting competitions organized by CDC (Reich et al., 2019). The CDC defines the onset as the week at the past three consecutive weeks that have ILI above a baseline value defined by CDC. For US dataset, these values are usually close to 2.2. Since the Japanese data does not have onset baselines, we only perform peak time and intensity prediction. For each of the baselines, we append a classification head on top of the aggregated output embeddings for peak and onset week prediction and train using cross-entropy loss. For peak intensity prediction, we append a final layer that outputs a single scalar value and train using MSE loss. The results are summarized in Table 2. We observe that PEM outperforms all baselines in most of the tasks and is comparable to the top-performing baseline for Peak intensity prediction in Influenza-Japan. This shows that PEM can outperform the baselines in many epidemic analysis tasks across multiple diseases and geographical locations.

Table 2: Peak and onset prediction performance of PEM and baselines. Top performing scores are in **bold** and the second best is underlined.

| Datasets | Task | General-time Series | | | | | | Epi-specific | | | PEM |
|---|---|---|---|---|---|---|---|---|---|---|---|
| | | AF | IF | PT | DL | TN | MICN | EF | ED | EB | |
| **Flu-US** | Peak week | 6.37 | 7.33 | 8.38 | 8.91 | 7.32 | 7.15 | 6.92 | 6.3 | 5.22 | **5.18** |
| | Peak intensity | 0.94 | 0.93 | 1.13 | 1.14 | 1.27 | 1.03 | 0.85 | 0.97 | 1.05 | **0.72** |
| | Onset week | 6.49 | 8.33 | 9.17 | 9.34 | 7.41 | 7.55 | 7.26 | 6.11 | **5.28** | 6.11 |
| **Flu-Japan** | Peak week | 6.49 | 6.44 | 8.14 | 8.52 | 8.21 | 7.83 | 6.44 | 5.19 | 4.97 | **4.72** |
| | Peak intensity | 983 | 1066 | 1289 | 1187 | 1052 | 974 | 915 | 874 | 1046 | **864** |

**Effect of pre-training on efficient fine-tuning (Q2)**  The time taken to train *till convergence* and memory requirements for PEM is similar to the transformer-based baselines. PEM takes 47-88% of training time taken by state-of-art baselines to reach similar performance. This shows that pre-training enables both *faster as well as more effective training*. See Appendix §C for details.

**Data efficiency of PEM (Q3)**  Pre-training enables large language models to adapt ot downstream tasks using small amount of data (Brown et al., 2020). We measured the performance of PEM using various fractions of training data and observed that PEM typically equals or outperforms best baselines using 60-80% of training data (See Appendix Figure 4d).

**Generalization to novel diseases (Q4)**  In real-world scenarios we may encounter novel epidemic dynamics not seen in pre-train datasets such as outbreak of a novel disease or deploying the model to novel geographical location. Therefore, we measure the efficacy of PEM when we remove the disease in downstream task from pre-training data. PEM provides state-of-art performance in most cases. We also show that PEM can *adapt to the dynamics of the novel Covid-19 pandemic*, which is not present during pre-training by providing 2-12% better performance over previous state-of-art models. The detailed results are discussed in Appendix §D.

**Comparison with alternate SSL methods (Q5)**  Using SSL to improve representation learning of time-series has been explored in prior works. However, the goal of these methods is narrow: they only aim to improve performance for a specific task by performing SSL on the same training dataset. Therefore, these methods may not efficiently learn from multiple disease datasets and generalize well to a wide range of downstream tasks. We compare PEM with these previous works: TS2Vec (Yue et al., 2022), TNC (Tonekaboni et al., 2021) and TCC (Eldele et al., 2021). We use these methods to train on all of pre-train datasets $\mathcal{D}_{pre}$ and fine-tune the models for each of the tasks by appending the appropriate output layer. We also measure the impact of each of the SSL tasks $\mathcal{T}_{pre}$ on all the benchmarks. Finally, we also examine the impact of linear probing and pre-training as a whole.

The results are summarized in Table 3. PEM outperforms the alternate SSL methods that fail to even beat top baselines in most cases. We see over 15% improvement in all forecasting tasks, over 31% improvement in peak week prediction, and 27% improvement in peak intensity prediction.

Table 3: Comparison of PEM with alternate SSL methods and pre-training with each of the SSL tasks in $\mathcal{T}_{\text{pre}}$ independently.

| | Forecasting | | | | Peak week | | Peak intensity | |
|---|---|---|---|---|---|---|---|---|
| Model | Flu-US | Flu-japan | Crypto. | Typhoid | Flu-US | Flu-japan | Flu-US | Flu-Japan |
| SSL Methods | | | | | | | | |
| TS2Vec | 1.85 | 1175.3 | 247.6 | 6.11 | 7.33 | 7.22 | 0.95 | 1198 |
| TNC | 1.22 | 1059.6 | 317.4 | 7.92 | 8.18 | 6.91 | 0.83 | 1045 |
| TS-TCC | 1.94 | 1344.6 | 306.6 | 5.68 | 7.94 | 6.88 | 1.05 | 1079 |
| Ablation variants | | | | | | | | |
| No Pre-training | 0.85 | 1577.3 | 285 | 5.61 | 8.38 | 8.14 | 1.13 | 1287 |
| No Linear Probing | 0.51 | 979.5 | 211.5 | 4.06 | **5.12** | 4.88 | **0.68** | 882 |
| Only RANDMASK | 0.85 | 1473.5 | 238.2 | 5.11 | 9.22 | 6.94 | 0.98 | 1055 |
| Only PEAKMASK | 0.63 | 1244.7 | 238.7 | 5.19 | 5.82 | 5.27 | 0.82 | 885 |
| Only LASTMASK | 0.72 | 1129.5 | 222.4 | 4.17 | 6.11 | 6.25 | 0.96 | 1074 |
| Only SeasonSelect | 0.71 | 1443.6 | 227.3 | 5.29 | 7.3 | 7.19 | 0.93 | 917 |
| PEM | **0.50** | **957.2** | **192** | **3.76** | 5.18 | **4.72** | 0.72 | **864** |

With regard to the impact of each of the tasks in $\mathcal{T}_{\text{pre}}$, we observe that PEM pre-trained with all of the tasks performs better those pre-trained with any single SSL tasks. For forecasting tasks, we find that PEAKMASK or LASTMASK are the most important SSL methods. We also observe that two-step pre-training helps improve downstream performance and without any pre-training the model performance deteriorates to a large extent, significantly underperforming many of the baselines.

**Modeling choices and Hyperparameter sensitivity (Q6)**   We also study the impact of various model design choices and the sensitivity of important hyperparameters. The detailed results of the study are presented in Appendix §E. We studied the impact of two important model design choices: segmentation (§3.1) and instance normalization (§3.3). We observed a 27-75% decrease in performance by tokenizing individual time-steps instead of segments and an 8-31% decrease in performance without instance normalization. We studied the effect of SSL hyperparameters and segment size. The hyperparameters of the segment size ($P = 4$), $\gamma = 0.2$ for RANDMASK and $\gamma = 0.1$ for LASTMASK generally perform the best if not close to best across multiple diseases.

## 6   CONCLUSION

We study the challenge of leveraging heterogeneous epidemic time-series data across multiple sources via pre-trained models that can be fine-tuned to outperform previous state-of-the-art models in diverse epidemic analysis tasks concerning a wide range of diseases. PEM provided 11-24% better forecasting performance and 9-12% improvement in peak prediction outperforming strong general time-series forecasting and epidemic-specific forecasting models. We also showed the importance of specific modeling choices such as segmenting input time-series (Nie et al., 2022) as individual tokens and more importantly, designing effective SSL tasks that can learn from cross-disease datasets of multiple diseases. Our work is the first to show the efficacy of pre-training on a wide array of unlabeled datasets from multiple unrelated sources as a viable method to improve model performance across multiple applications. While our backbone architecture is fairly straightforward, our SSL methods can be easily used to extend a wide range of model architectures that ingest time-series data. Our work could also lead to further important research in the direction of general pre-trained models for time-series similar to pre-trained models in language and vision domains. Our approach can be extended to other applications such as health care, economics, and sales which typically have a large number of datasets from multiple sources. We can also easily extend to other time-series tasks such as classification and anomaly detection.

Our work is limited to time-series data whereas other multimodal data epidemic sources may also be available for diseases such as mobility networks, geographical relations, social media, etc. that cannot be directly integrated into PEM. Adapting PEM to leverage these features of varying temporal scales (Rodríguez et al., 2022b; Ibrahim et al., 2021) and providing probabilistic forecasts with uncertainty quantification (Xu & Xie, 2021; Kamarthi et al., 2021) are important research directions. Due to our method's relevance to critical public health applications and decision-making, the potential misuse of our model can not be discounted. Steps should be taken to alleviate problems such as disparities in the quality of data collected across regions, equity of prediction performance, etc.

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
