# Appendix for PEMs: Pre-trained Epidemic Time-Series Models

## A    RELATED WORKS

**Neural models for time-series analysis**    Deep neural networks have been widely used in many time series forecasting applications with great success. DeepAR Salinas et al. (2020) is a popular forecasting model that trains an auto-regressive recurrent network to predict the parameters of the forecast distributions. Other works including deep Markov models Krishnan et al. (2017) and deep state space models Rangapuram et al. (2018); Li et al. (2021); Gu et al. (2021) explicitly model the transition and emission components with neural networks. Recent works have also leveraged transformer-based models, which have been widely used for language modeling, on general time-series forecasting Oreshkin et al. (2019). Other works have extended the transformer architecture to improve efficiency and better capture long-term temporal trends resulting in state-of-art performance in many long-term forecasting benchmarks Zhou et al. (2021); Chen et al. (2021); Zhou et al. (2022); Liu et al. (2021). However, all these methods do not leverage pre-training. They follow the typical supervised learning paradigm of leveraging training data from past values of the same dataset to forecast future values and do not leverage cross-domain heterogenous datasets or aim to provide generalized models that can be used for a wide range of heterogeneous tasks.

**Self-supervised learning for time-series**    Recent works have shown the efficacy of self-supervised representation learning for time-series for various classification and forecasting tasks in wide range of applications such as modeling behavioral datasets Merrill & Althoff (2022); Chowdhury et al. (2022), power generation Zhang et al. (2019), health care Zhang et al. (2022). Franceschi et al. (2019) used triplet loss to discriminate segments of the same time-series from others. TS-TCC used contrastive loss with different augmentations of time-series Eldele et al. (2021). TNC Tonekaboni et al. (2021) use the idea of leveraging neighborhood similarity for unsupervised learning of local distribution of temporal dynamics. TS2Vec leveraged hierarchical contrastive loss across multiple scales of the time-series Yue et al. (2022). However, all these methods apply SSL on the same dataset that is used for training and may not adapt well to using time-series multiple sources such as time-series from multiple diseases. Our work, in contrast, tackles the problem of learning general models from a wide range of heterogenous datasets that can be fine-tuned for a wide variety of tasks on multiple datasets that may not be used during pre-training. Therefore, we design SSL tasks that can adapt to multiple time-series datasets and capture useful underlying properties from these datasets for superior performance on multiple downstream applications on various disease forecasting tasks.

**Statistical models for epidemic forecasting**    Due to recent advances in machine learning and deep learning as well as the availability of datasets from various surveillance sources, statistical and deep-learning-based models are increasingly used for epidemic forecasting tasks with great success Rodríguez et al. (2022b). Classical auto-regressive time-series models like ARIMA and its variants have been adapted for disease forecasting Soebiyanto et al. (2010); Yang et al. (2015). Other models use Bayesian generative approach Brooks et al. (2015; 2018) to provide probabilistic forecasts and have been successful in past epidemic forecasting competitions like Flusight Reich et al. (2019). Other classical machine-learning methods like Gaussian Processes Zimmer & Yaesoubi (2020), Generalized Linear models Chakraborty et al. (2018) and nearest-neighbor-based regression Chakraborty et al. (2014) have also been adapted.

Recent works have also used deep learning-based methods that are flexible to various data sources and capture complex temporal patterns. While some use off-the-shelf recurrent neural models Venna et al. (2018), others exploit important characteristics of epidemic dynamics such as dynamically modeling sequence similarity across seasons Adhikari et al. (2019) and uncertainty with past seasons Kamarthi et al. (2021), exploiting spatial relations Deng et al. (2020); Kamarthi et al. (2022) as well as leveraging priors from traditional mechanistic models Rodríguez et al. (2022a); Gao et al. (2021). However, most previous works train only from past data for epidemics they forecast and do not leverage useful background knowledge from a large amount of epidemic data of other diseases collected in the past.

# B    ADDDITIONAL DETAILS ON MODEL ARCHITECTURE AND TRAINING

We use a 6-layer transformer encoder with 8 attention heads each for PEM. For all pre-training and all downstream tasks, we set the segment size $P = 4$ and $\gamma$ as 0.2 for RANDMASK and 0.1 for LASTMASK tasks. We use learning rate of $10^{-4}$ for pre-training on all SSL tasks and during training simultaneously and use early stopping for training, training to a maximum of 5000 epochs. We found that pre-training for up to 5000 epochs on all SSL tasks simultaneously was sufficient, as longer pre-training did not improve SSL-related losses or downstream performance significantly. During training, we set 5000 epochs as the maximum, but we observed that most downstream tasks required 1500-2500 epochs to converge and reach the early stopping criteria. Since the datasets in most tasks could fit into the GPU, we set the batch size to be equal to the number of training data points.

The models were trained on Nvidia Tesla V100 GPU. We also provide a link to anonymized code and datasets[4].

# C    TRAINING TIME AND MEMORY

We compare the average training time till convergence and memory used by PEM and baselines in Table 4. We observe that the training time and memory consumption of PEM is similar to neural baselines while providing significantly more accurate forecasts. Note that FUNNEL and EB are non-deel learning statistical models that use lower parameters and hence use significantly less training time and memory but provide worse performance.

Table 4: Average training time and maximum memory taken by each of the baselines and PEM for each disease.

| Model/Benchmark | Average Training time(min) | | | | Max. Memory(GB) | | | |
|---|---|---|---|---|---|---|---|---|
| | Flu-US | Flu-Japan | Crypto. | Typhoid | Flu-US | Flu-Japan | Crypto. | Typhoid |
| AF | 37.9 | 31.6 | 29.7 | 49.5 | 4.2 | 3.8 | 4.9 | 3.7 |
| IF | 31.6 | 42.5 | 35.9 | 55.1 | 4.5 | 3.7 | 4.3 | 3.2 |
| PT | 46.7 | 41.3 | 44.8 | 41.2 | 4.7 | 3.5 | 4.8 | 4.1 |
| DL | 32.5 | 31.7 | 31.6 | 47.2 | 3.2 | 3.7 | 3.7 | 3.5 |
| TN | 42.7 | 37.5 | 39.1 | 51.7 | 4.3 | 4.7 | 4.2 | 4.3 |
| MICN | 36.3 | 39.2 | 36.4 | 48.1 | 3.1 | 3.2 | 3.7 | 3.2 |
| EF | 27.4 | 22.5 | 29.3 | 47.2 | 2.8 | 2.1 | 3.5 | 3.1 |
| ED | 39.1 | 42.7 | 39.6 | 53.6 | 3.2 | 2.7 | 3.4 | 3.1 |
| EB | 3.4 | 3.2 | 3.9 | 3.5 | 0.1 | 0.1 | 0.1 | 0.1 |
| FUNNEL | 0.6 | 0.5 | 0.9 | 0.2 | 0.1 | 0.1 | 0.13 | 0.1 |
| PEM | 35.4 | 25.5 | 29.2 | 64.5 | 4.7 | 3.5 | 4.8 | 4.1 |

Further, we measure the average training time taken by PEM to match the forecast RMSE of the baselines in Table 5. We observe that PEM matches previous state-of-art performance in much less training time before beating it when trained to convergence.

Table 5: Comparison of training time taken by PEM to match the performance of the best-performing baseline for each benchmark.

| | Flu-US | Flu-Japan | Cryptosporidiodia | Typhoid |
|---|---|---|---|---|
| Avg. training time taken to reach performance of best baseline | 19.5 | 20.2 | 23.7 | 25.9 |
| TIme taken by best baseline | 27.4 | 22.5 | 29.3 | 48.1 |

---

[4]Anonymized code link: `https://anonymous.4open.science/r/EmbedTS-3F5D/`

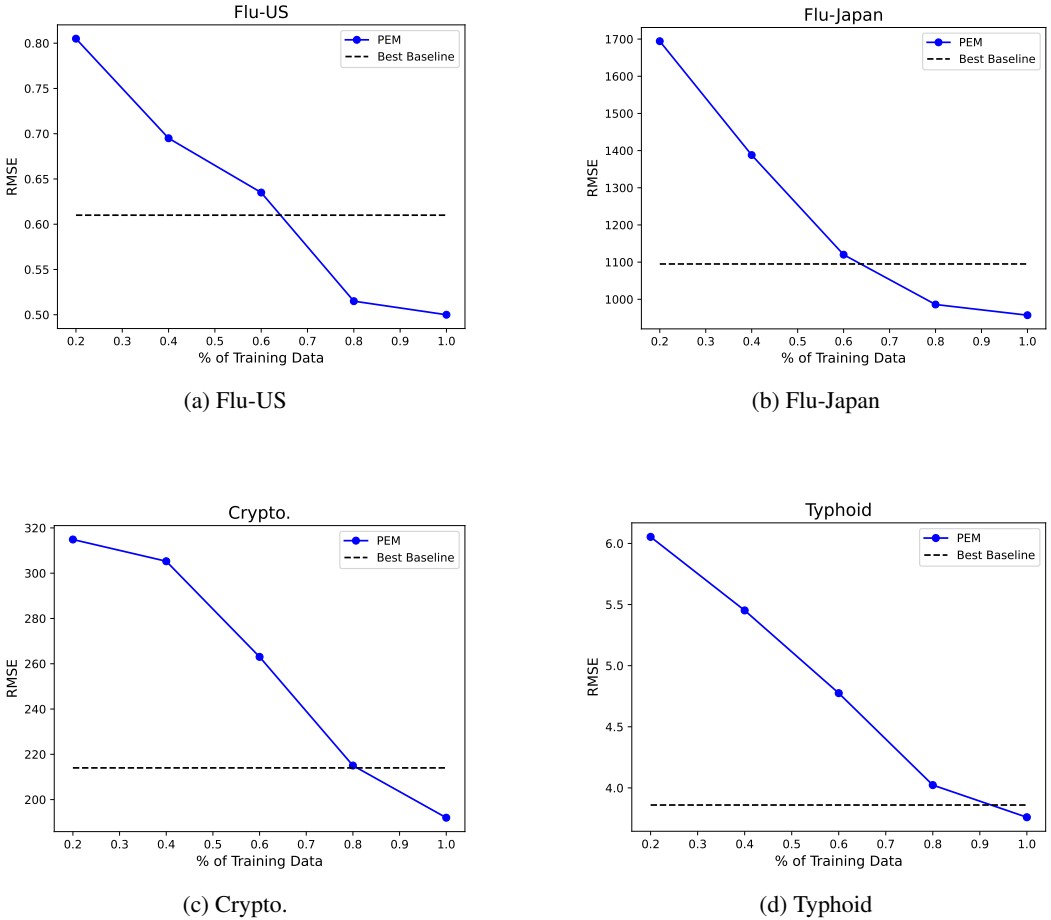

(a) Flu-US

(b) Flu-Japan

(c) Crypto.

(d) Typhoid

Figure 4: Performance of PEM with varying fractions of training data. Performance in averaged over 5 runs. Note that in most cases PEM's performance is superior to best baseline using less than 80% of data.

# D ADAPTING TO UNSEEN DISEASES DURING PRE-TRAINING (Q2)

One of the important goals of pre-training on a large number of multi-domain disease datasets is to capture underlying patterns and information that are observed across time-series of multiple diseases that can be generalized to newer training datasets as well as previously unseen diseases during pre-training. The diseases considered in Section 5 had past data used during pre-training. In this section, we evaluate how well PEM adapts to scenarios where the disease of the training dataset is not used during pre-training.

Table 6: Comparison of forecasting performance (RMSE) of PEM removing the disease used for training for pre-training with the original PEM and performance of the best baseline.

| Dataset | Best Baseline | PEM | PEM-ExcludeTrain |
|---|---|---|---|
| Influenza-US | 0.62 | 0.5 | 0.61 |
| Influenza-Japan | 1466 | 957.2 | 997.6 |
| Cryptosporidosis | 214 | 192 | 217.8 |
| Typhoid | 4.67 | 3.76 | 4.58 |

**Forecasting on unseen diseases** For each of the training tasks, we pre-train PEM removing the disease used for training from $\mathcal{D}_{pre}$. We call this version of PEM as PEM-ExcludeTrain. We compare PEM-ExcludeTrain with PEM and baselines in Table 6. While PEM-ExcludeTrain's performance is worse compared to PEM, in most cases its performance is comparable to if not better than the best baseline for each of the forecasting tasks.

Table 7: Forecasting performance on the previously unseen Covid-19 mortality in US from June 2020 to June 2021.

| Week ahead | AF | IF | PT | DL | TN | MICN | EF | ED | EB | PEM |
|---|---|---|---|---|---|---|---|---|---|---|
| 1 | 36.3 | **25.2** | 31.6 | **26.1** | 29.3 | 27.4 | 32.7 | 48.2 | 45.2 | 29.7 |
| 2 | 44.5 | **37.1** | 42.7 | 42.4 | 44.7 | 41.5 | 38.9 | 53.2 | 49.7 | 38.4 |
| 3 | 59.3 | 69.2 | 55.2 | 56.9 | 59.1 | 54.7 | 53.7 | 79.3 | 73.4 | **48.6** |
| 4 | 66.2 | 84.7 | 59.1 | 59.2 | 63.3 | 59.1 | 68.2 | 81.4 | 85.9 | **52.6** |
| Avg | 51.6 | 54.1 | 47.2 | 46.2 | 49.1 | 45.7 | 48.4 | 65.5 | 63.6 | **42.3** |

**Case-study on Covid-19** We further provide a realistic case study to illustrate the importance of adapting to unseen diseases from pre-training by evaluating the performance of PEM and baselines on the novel Covid-19 pandemic. We focus on forecasting weekly mortality from Covid-19 in the US Cramer et al. (2022). We do not use any Covid-19 related data in $\mathcal{D}_{pre}$ and only use past Covid-19 data for training PEM for each prediction week via the real-time forecasting setup similar to Section 5. The results are summarized in Table 7. On average, we observe a 2% improvement in forecasting performance over the best baseline with respectable 4% and 12% improvement in harder three and four-week ahead forecasts. Therefore, PEM can successfully leverage pre-training to adapt to even unseen novel pandemics like Covid-19.

# E ABLATION STUDIES (Q4)

In this section, we study the impact of various model design choices on the performance of PEM as well as the parameter sensitivity of some important hyperparameters of PEM.

**Importance of segmentation and reversible instance normalization** The superior performance of PEM is the result of various design choices related to model architecture as well as pre-training methods. We studied the impact of each of the SSL tasks in Section 5. Here, we observe the impact of important architectural choices of PEM on top of the transformer architecture: using segmentation and instance normalization Kim et al. (2021). Segments of input time-series are used as tokens instead of individual time-stamps to provide a better semantic representation of the temporal locality of the time-series. We use reversible instance normalization to accommodate time-series of various magnitudes as well as provide robustness against the distributional shift in individual time-series data.

Table 8: Ablation study of the impact of SSL, segmentation and normalization on PEM performance.

| Task | Disease | PEM-No Segments | PEM-No Reversible Norm. | PEM |
|---|---|---|---|---|
| Forecasting | Flu-US | 0.96 | 0.54 | **0.5** |
| | Flu-Japan | 1373.7 | 10165 | **957.2** |
| | Crypto. | 257.2 | 229.4 | **192** |
| | Typhoid | 4.81 | 4.16 | **3.76** |
| Peak week | Flu-US | 7.26 | 5.39 | **5.18** |
| | Flu-Japan | 6.33 | 6.39 | **4.72** |
| Peak intensity | Flu-US | 0.81 | 0.95 | **0.72** |
| | Flu-Japan | 1197 | 1083 | **864** |

The ablation study is summarized in Table 8. First, we observe that PEM with both components performs better than its ablation variants. We also observe that without segmentation, the performance decreases by about 75% in forecasting, 35% in peak week prediction and 27% in peak intensity prediction, underperforming many baselines. Finally, using reversible instance normalization has the most impact on peak intensity prediction at 31% whereas only decreases forecasting performance by about 8%. Therefore, reversible instance normalization helps adapt to and model data around the peaks which can cause distributional shifts in time-series.

Table 9: Influence of important hyperparameters on average forecasting performance. The default hyperparameter values are underlined.

| Hyperparameter | Value | Flu-US | Flu-Japan | Cryptosporidiosia | Typhoid |
|---|---|---|---|---|---|
| Segment size | 2 | 0.79 | 1366.8 | 247.4 | 5.77 |
| | 4 | **0.5** | **957.2** | **192** | **3.76** |
| | 8 | 0.59 | 996.2 | 229.8 | 4.69 |
| RANDMASK $\gamma$ | 0.1 | 0.55 | 973.7 | 219.5 | 4.13 |
| | 0.2 | **0.5** | **957.2** | **192** | **3.76** |
| | 0.4 | 0.62 | 1079.5 | 286.9 | 6.05 |
| LASTMASK $\gamma$ | 0.1 | **0.5** | **957.2** | 192 | 3.76 |
| | 0.2 | 0.53 | 1026.8 | **186.3** | **3.51** |
| | 0.4 | 0.68 | 1277.5 | 287.2 | 5.37 |

**Hyperparameter sensitivity analysis** We also study important hyperparameters of PEM on performance in downstream forecasting tasks. We vary the length of the segments $P$ of the input time-series as well as tune the hardness of the SSL tasks RANDMASK and LASTMASK by tuning the value of $\gamma$ for each of the tasks. The average forecasting performance is summarized in Table 9. We observe that the default hyperparameters of the segment size ($P = 4$), $\gamma = 0.2$ for RANDMASK and $\gamma = 0.1$ for LASTMASK generally perform the best if not close to best across multiple diseases. Therefore, the important *hyperparameters are not sensitive to specific downstream tasks*. We also observe that increasing $\gamma$ to a higher value of 0.4 quickly degrades the performance in general since the reconstruction task gets increasingly harder with an increase in $\gamma$.