# OpenReview forum: "PEMs: Pre-trained Epidemic Time-Series Models"
_ICLR.cc/2024/Conference — ICLR 2024 Conference Withdrawn Submission_

### Official Review · Reviewer_7guK · 2023-10-31

**Soundness:** 2 fair
**Presentation:** 3 good
**Contribution:** 2 fair
**Rating:** 3
**Confidence:** 5

**Summary:**

The authors propose Pre-trained Epidemic Models (PEMs) that is capable of learning pattern across multiple datasets and multiple diseases using self-supervised learning.  They demonstrate the success of their method in the task of forecasting. Their approach maintains or even
surpasses the performance of other strong deep-learning models.

**Strengths:**

1. The paper is well-presented and easy to understand.
1. PEM is a novel model that uses an array of self-supervised learning (SSL) tasks shown to be effective in learning from cross-disease datasets.
1. PEM demonstrates efficiency in utilizing multiple training datasets while maintaining or even surpassing the performance of other strong deep learning models. This showcases its ability to adapt to downstream tasks with limited data, which is particularly beneficial in real-world
scenarios where complete datasets may not be readily available, especially in the case of unseen diseases.

**Weaknesses:**

1. __Limitations in Model Compared__: While the paper compares PEM with FUNNEL, a mechanistic model, it does not explore simpler disease-focused mechanistic models or even basic auto-regressive models. Additionally, the absence of testing the performance of simpler models on unseen diseases, like COVID-19, limits the model's generalization claims. Over the years many experts have submitted forecasts for COVID-19 and influenza in real-time (e.g., US COVID-19 Forecast Hub, EU COVID-19 Forecast Hub). The paper already refers to a paper on one of the hubs (Cramer 2022). Without comparison against the models used by the community, it is hard to judge how much value the proposed approach adds. Further, for COVID-19 and epidemiological data in general, there is frequent data revision and backcorrection. The data that is available today is a much cleaner version compared to what was available at the time a forecast was made. Especially for COVID-19 the data revision history is available, therefore, the authors should test the dataset with the appropriate version, not the latest version. It is not clear if the authors have done so.

 2. **Data Preprocessing Details**: The paper lacks information on the data preprocessing steps performed, such as data smoothing and temporal alignment. Preprocessing often has a significant impact on real-time forecasting, especially, when there is a lot of backcorrection.

 3. **Ablation Studies**:
    1. Performance compared to simple transfer learning or doing some training on a single disease dataset (such as pre-training on just past flu data if the disease of interest is flu) would have been good. Randmask does not perform nearly as well as the others, so does adding this into the combination of tasks contribute meaningfully?
    1. The choice of hyperparameters matches the best-performing ones in the "ablation studies". Can the authors verify that the hyperparameter selection was done on a validation set and not on the test set?

1. **Additional Comments**:
    1. Lower RMSE Values for Flu in the US: The remarkably low RMSE values for Flu in the US  looks unusual. Some more context or explanations for these results would be useful. Was this normalized somehow (like per 100k)?

    1. Consolidation of Appendix Results: While it's useful to have supplementary information in the appendix, the paper might benefit from incorporating key results from Q2, Q3, Q4, and Q6 into the main text to provide a more comprehensive overview of the model's capabilities.

    1. Presentation issues: While easy the writing is easy to understand, the paper has many typos; it will benefit from a pass for presentation.

**Questions:**

Addressing the following, especially the first one, will significantly improve my evaluation:

1. Have the authors compared their results against models that were used by the epidemic forecasting community during COVID-19 and ILI forecasting tasks? (These forecasts are publicly available)

1. Was there any data pre-processing involved? How does that impact the results?

1. Can the authors verify that the hyperparameter selection was done on a validation set and not on the test set?

1. Why is RMSE for the US so low?

---

> ### Author Response · Authors · 2023-11-22
> **Response to Reviewer 7guK**
>
> **While the paper compares PEM with FUNNEL, a mechanistic model, it does not explore simpler disease-focused mechanistic models or even basic auto-regressive models....COVID-19, limits the model's generalization claims**
>
> We do compare against state-of-art epidemic forecasting models that have shown to beat past simpler statistical models on epidemic forecasting tasks and
> have won past epidemic forecasting competition[1,2,3].
> Therefore, we believe this is a fair comparison to provide evidence
> that PEM indeed provides state-of-art forecasting performance.
>
>
>
> **The paper already refers to a paper on one of the hubs (Cramer 2022). Without comparison against the models used by the community, it is hard to judge how much value the proposed approach adds.**
>
> There are two main reasons why direct comparisons with the models used
> by forecast hub is not feasible.
> First, our model only uses past values of the time-series to forecast future dynamics. It does not ingest additional external variables.
> Therefore, all the benchmarks are univariate time-series forecasting
> or prediction tasks and we evaluate state-of-art baselines designed for such tasks.
> Further, most models in the forecasting hubs used additional data sources and the models and datasets used are not public in most cases.
>
>
> **The data that is available today is a much cleaner version compared to what was available at the time a forecast was made. Especially for COVID-19 the data revision history is available, therefore, the authors should test the dataset with the appropriate version, not the latest version. It is not clear if the authors have done so.**
>
> All the baselines and the model are evaluated on the latest available version of
> Covid-19 mortality data for a fair comparison.
> We agree with the reviewer that data revision is an important
> problem in epidemic forecasting. Our work doesn't address this problem and instead focuses on modeling and accurately forecasting
> epidemic time-series. However, analyzing and tackling the data revision
> problem for our pre-training framework is an interesting future direction.
>
>
> **The paper lacks information on the data preprocessing...**
>
> We do not perform any data pre-processing on raw time-series data provided. Since we used Reversible Instance Normalization  as part of the model, we
> do not need explicit data normalization before feeding into the model.
>
>
> **Performance compared to simple transfer learning or doing some training on a single disease dataset (such as pre-training on just past flu data if the disease of interest is flu) would have been good.**
>
>
> **Randmask does not perform nearly as well as the others, so does adding this into the combination of tasks contribute meaningfully?**
>
> Yes RANDMASK alone is the worse performing SSL task overall. However,
> we still included it for pre-training because (a) it still outperforms
> the variant without any pre-training, (b) and using all four SSL tasks
> together provided the best performance across all benchmarks.
>
>
> **Can the authors verify that the hyperparameter selection was done on a validation set and not on the test set?**
>
> Yes, the hyperparameter selection was done during training on validation data for all benchmarks.
>
>
> **Lower RMSE Values for Flu in the US**
>
> The influenza incidence time-series extracted from CDC is normalized
> between 0 to around 10. We direcltly use these values for training and prediction. Therefore, the RMSE values may appear very low compared to other benchmarks.
>
>
> *References*
>
> [1] Harshavardhan Kamarthi, Lingkai Kong, Alexander Rodríguez, Chao Zhang, and B Aditya Prakash.
> When in doubt: Neural non-parametric uncertainty quantification for epidemic forecasting. Advances in Neural Information Processing Systems
>
> [2] Bijaya Adhikari, Xinfeng Xu, Naren Ramakrishnan, and B Aditya Prakash. Epideep: Exploiting
> embeddings for epidemic forecasting. In Proceedings of the 25th ACM SIGKDD international
> conference on knowledge discovery & data mining
>
> [3] Logan C Brooks, David C Farrow, Sangwon Hyun, Ryan J Tibshirani, and Roni Rosenfeld. Flexible
> modeling of epidemics with an empirical bayes framework. PLoS computational biology,

---

### Official Review · Reviewer_A18H · 2023-10-31

**Soundness:** 2 fair
**Presentation:** 3 good
**Contribution:** 2 fair
**Rating:** 5
**Confidence:** 3

**Summary:**

- novel application of SSL and fine-tuning on broad epidemic data to get PEMs, which they use to forecast various diseases, as well as various other epidemic dynamics such as peak weeks and onset weeks.

- interesting experiments of generalization to novel diseases in appendix D

**Strengths:**

- originality: This is a new application domain for SSL as far as I know, but the methodology is not particularly new.

- quality: The submission is well written and formatted, with strong experimental results and ablations.

- clarity: most of the training details are there, though much is relegated to the appendix.

- significance: the work is significant as an application of SSL to a specific domain with greater performance, and finds relevance in the current post-pandemic context.

**Weaknesses:**

- sounds like your segments of time series is basically patching, similar to (Nie et al. 2022). If so, why do you achieve so much better results than PatchTST? Is it just the finetuning?

- I'm not sure I understand why inputting each time step as a single token would lack "semantic meaning", seeing as the output representations of a transformer for a given token are influenced by all the other tokens in the input sequence (i.e. "contextual embeddings")

- How do you normalize the training process of your method wrt baselines? How can you disentangle the architecture's influence from the influence of the training process and the influence of using additional compute?

- unclear what is being evaluated in appendix D table 6 since it's not known how the baselines are trained. For example, it is clearly remarked that no pre-training is performed for the PT baseline, but not why.

- why are the results around one of the contributions (Significant improvement in data and training efficiency and adaptability to novel epidemics) only in the appendix?

**Questions:**

- There might be bleed between the influenza data from 2001 to 2010 and the crypto data from 2006 to 2012 given the temporal alignment, but I'm not knowledgeable enough to judge how strong the overlap could be. Is the argument that, since influenza and crypto have different inherent dynamics, they shouldn't influence each other?

- How do you choose the value of T, i.e. the length of the input time series?

- How do you justify the default choice of gamma = 0.1 for lastmask ? Seems like 0.2 performs well too.

Overall, I am rating this a 5 (marginally below acceptance) before discussion, since it is hard to disentangle the impact of the supervised pre-training from the architectural design choices and the increased compute/data. There are no details on how the baselines are trained, so these comparisons do not properly motivate which components of PEM lead to its superior performance. I am very amenable to changing this rating given discussions around how the baselines were implemented for fair comparisons.

---

> ### Author Response · Authors · 2023-11-22
> **Response to Reviewer A18H**
>
> **why do you achieve so much better results than PatchTST? Is it just the finetuning?**
>
> It is due to our SSL pre-training. Indeed performing fine-tuning on heterogenous time-series data by designing specific SSL tasks for extracting useful background knowledge from wide range of disease datasets during pre-training is our most important technical contribution.
> We note that most other SSL pre-training methods of past works are not effective
> on our heterogenous pre-training problem.
> Therefore, our method of effective fine-tuning enables the pre-trained models
> to significantly outperform models without pre-training (such as using PatchTST without pre-training) as well as methods that use
> previous SSL methods.
>
> **I'm not sure I understand why inputting each time step as a single token would lack "semantic meaning", seeing as the output representations of a transformer for a given token are influenced by all the other tokens in the input sequence (i.e. "contextual embeddings")**
>
> Providing semantically meaningful input tokens can better enable transformers to model
> relations between tokens.
> For example, in the context of language, LLMs use specific tokenization methods to capture semantic meaning instead of individual characters to provide significantly better performance.
> Further, the inputs of time-series are real-valued numbers rather than specific character.
> Therefore, these individual values of time-series observed in isolation do not provide sufficient context such as local patterns (such as peaks, spikes, etc.)
> around the time-step.
> therefore, we use segments of time-series that provide semantic information about local patterns.
>
> **How do you normalize the training process of your method wrt baselines? How can you disentangle the architecture's influence from the influence of the training process and the influence of using additional compute?**
>
> For all deep learning baselines and PEM, we train using early stopping.
> We use the same compute hardware for training all models and baselines (appendix B) and we measure the training time till convergence
> for all models.
> PEM uses similar training time to other best baselines with similar compute and provide significantly better performance.
> In fact, it takes 12-50% less time to match the performance of the baselines (Appendix Table 5).
>
> We also measure the importance of various training choices and pre-training choices in Table 3 and show the importance of each SSL task
> and linear-probing for optimal performance. We also measure the effect of segmentation in Appendix Table 9.
>
> **unclear what is being evaluated in appendix D table 6 since it's not known how the baselines are trained. For example, it is clearly remarked that no pre-training is performed for the PT baseline, but not why**
>
> Appendix D Table 6 measures the performance of PEM when we remove the
> training data for downstream tasks from pre-training.
> This is denoted by PEM-ExcludeTrain.
> This measures the performance of fine-tuning PEM to unseen diseases.
> We observe that even is such cases PEM outperforms the best baseline performance.
>
> **why are the results around one of the contributions (Significant improvement in data and training efficiency and adaptability to novel epidemics) only in the appendix?**
>
> We summarize the main results of training efficiency and data efficiency in the main paper page 8 and provide additional details in the Appendix: PEM takes 47-88% of the time taken by best baselines and uses 60-80% of total training data to outperform them.
> We could summarize the training times in Table 4 using the bar graph and add to the main paper.
>
> **There might be bleed between the influenza data from 2001 to 2010 and the crypto data from 2006 to 2012 given the temporal alignment**
>
> This is an interesting question. However, influenza and crypto show very different disease dynamics. Influenza peaks during winter months and spreads through the air. Crypto mostly occurs during the summer and is spread through food and water contamination.
> While influenza is spread through a virus, crypto is spread through a
> microscopic parasite.
> We couldn't find any research linking the incidence of both diseases
>
> **How do you choose the value of T, i.e. the length of the input time series?**
>
> All the diseases in training and pre-training have a sampling rate of 1 month. Moreover, even seasonal diseases have periodicity within 12 months.
> Therefore, we choose the total input length to be $T=12$.
> We will add this information in the Appendix.
>
> **How do you justify the default choice of gamma = 0.1 for lastmask ? Seems like 0.2 performs well too.**
>
> Yes both 0.2 and 0.1 performs well depends on the downstream tasks.
> However, even in cases 0.2 performs better, 0.1 is only slightly worse.
> Therefore, we chose gamma as 0.1.

---

### Official Review · Reviewer_ZKwe · 2023-11-02

**Soundness:** 3 good
**Presentation:** 3 good
**Contribution:** 2 fair
**Rating:** 6
**Confidence:** 4

**Summary:**

This paper proposes a pre-training machnism for using transformers to make time series forecasting for epidemics. The pre-training is self-supervised learning through multiple tasks: randmask, lastmask, peakmask, seasonselect. Similar to large language model pre-training, the proposed pre-training method for time series forecasting learn general patterns from multiple disease datasets. Then fine tune the pre-trained model to a specific disease forecasting task. The method is demonstrated on several real world epidemic forecasting tasks and outperforms the baselines.

**Strengths:**

The idea of pre-train a time-series epidemic forecasting model is very interesting and new. This has not been done before. The method is technically sound and the writing is clear. The experimental design is reasonable and results show superior performance compared with the baselines.

**Weaknesses:**

1. The baselines in this paper do not represent the SOTA performance. Also, the baselines are not "easy-to-replicate" methods whose code is not provide. This makes me a bit concern about the effectiveness of the comparison results.
2. There is no uncertainty quantification analysis of all the methods. It seems the transformer-based models are so large so may have large predicting variance.

**Questions:**

1. In section SEASONDETECT, the paper first mentions that the year is divided into 4 seasons Season 1 (Dec-Feb), but later, it also says the peak season is s_1(d). Could you clarify this in a more consistent way?
2. Data normalization, will the normalized data be reversed before computing the loss error during training?
3. Why perform linear probing before fully fine-tuning?
4. What's the pre-training cost?

**Details Of Ethics Concerns:**

The pre-training process involves multiple disease datasets, there may have bias introduced by a disease data due to the nature of disease characteristics, environmental factors, demographic factors, etc. There is no discussion about the possible bias in the pre-training process.

---

> ### Author Response · Authors · 2023-11-22
> **Response to Reviewer ZKwe**
>
> **The baselines in this paper do not represent the SOTA performance.**
>
> The chosen baselines are state-of-art general forecasting and epidemic forecasting
> models at the time of submission. Therefore, we are indeed comparing PEM with the previous best possible performance.
>
> The baselines used have code released publicly. We used the same code with
> mostly default hyperparameters for evaluation with hyperparameter tuning on learning rate.
> We can add links to the baseline code in paper to alleviate these concerns.
>
> **There is no uncertainty quantification analysis of all the methods. It seems the transformer-based models are so large that they may have large predicting variance**
>
> Our method and most other baselines only focuses on forecasting and prediction accuracy and does not
> provide reliable uncertainty estimates.
> However, we agree with the reviewer that uncertainty quantification is an important forecasting problem, and extending our methods to produce calibrated probabilistic forecasting is an interesting research direction.
>
> **Clarification on SEASONDETECT**
>
> We divide the 12 months into four seasons: Dec-Feb, Mar-May, June-Aug and Sept-Nov.
> For each disease dataset $d$, we assign one of these four seasons as peak season $s_1(d)$. For example, in the case of influenza $s_1(d) =$ Sept-Nov.
> Based on assignment on $s_1(d)$ we assign $s_2(d), s_3(d), s_4(d)$ chronologically.
> In influenza's case, $s_2(d)=$ Dec-Feb, $s_3(d)=$ Mar-May, $s_4(d)=$ June-Aug.
>
> **Data normalization, will the normalized data be reversed before computing the loss error during training?**
>
> Yes. Using Reversible Normalization prevents large gradients during training.
>
> **Why perform linear-probing before fine-tuning?**
>
> Kumar et. al [1] showed that doing direct fine-tuning leads to lower accuracy
> as it distorts the learned representations of pre-trained models, especially for downstream tasks with distribution shift, similar to our case where we fine-tune on heterogeneous and unseen disease datasets.
> Linear-probing alleviates this by adapting the final layer of the pre-trained model
> to downstream tasks without distorting the representation learning of the pre-trained model.

---

### Official Review · Reviewer_PcYs · 2023-11-05

**Soundness:** 3 good
**Presentation:** 3 good
**Contribution:** 2 fair
**Rating:** 5
**Confidence:** 3

**Summary:**

This paper aims to explore the pre-training of time series models for epidemic data. The proposed model is the first-ever pre-trained time series model for epidemic analysis tasks. Unlike SSL for images or texts, this paper presents four SSL tasks to capture various aspects of epidemic dynamics. Extensive experiments over real-world epidemic datasets verify the effectiveness of the proposed model.

**Strengths:**

1. The paper addresses an essential problem in time series domain.
2. The paper conducts extensive experiments to verify the effectiveness of PEM.
3. The experiments have verified the effectiveness of PEM, especially in comparison with those without SSL.
4. The experiments also cover a discussion over data efficiency and generalization to novel diseases.

**Weaknesses:**

1. The related work should be expanded. To the best of my knowledge, a series of papers [1] have been studied on SSL for general time series, also including epidemic analysis. However, they're rarely mentioned or considered as baselines for a fair comparison.
2. The technical novelty against previous SSL approaches is somewhat limited.
3. The baselines listed on Page 7 are insufficient. More SSL methods, particularly for general time series, should be included as baselines. For example, it would be interesting to see if PEM can outperform recently developed SSL methods in time series analysis.
4. It would greatly enhance reproducibility if the authors provide the source code for their approach.
5. I highly recommend that this paper provide more results on few-shot or zero-shot learning for epidemic data, as it is typically characterized by sparse data.

Reference:

[1] Zhang, Kexin, et al. "Self-Supervised Learning for Time Series Analysis: Taxonomy, Progress, and Prospects." arXiv preprint arXiv:2306.10125 (2023).

**Questions:**

If I have misunderstandings in the weaknesses, please clarify them in the rebuttal phase. Thank you.

---

> ### Author Response · Authors · 2023-11-22
> **Response to Reviewer PcYs**
>
> **The related work should be expanded.**
>
> We have described recent state-of-art SSL methods and used them as baselines.
> While there are many methods that perform SSL on time-series, we focus
> on methods that are applicable to any general time-series domain with
> specific applications to forecasting tasks.
> We would gladly add additional references to SSL methods including recent methods
> described in the survey referred to by the reviewer.
>
> **The technical novelty against previous SSL approaches is somewhat limited.**
>
>  Unlike most previous SSL papers, we are the first to deal with a different broader problem of effectively leveraging multiple heterogeneous epidemic datasets to train a general pre-train model that can be fine-tuned to a wide range of downstream epidemic analysis tasks.
>
> RANDMASK and LASTMASK are indeed widely used pre-training tasks in language domains and are also explored in time-series domain previously.
> However, to better extract useful epidemic dynamic information, we also introduce two novel tasks: PEAKMASK and SEASONDETECT. PEAKMASK helps identify peaks and their dynamics which is vital for epidemic forecasting since it is important to model and predict epidemic peaks in advance. Detecting seasonal information SEASONDETECT automatically is also important in successfully modeling epidemic dynamics. These tasks are carefully designed to impart useful background knowledge and patterns from multiple epidemic datasets.
>
> To further illustrate the importance of designing and choosing the right SSL tasks, we also compared PEM against sophisticated SOTA self-supervised methods for general time-series and show that using generic tasks that perform well in other domains to epidemic datasets leads to poor performance.
>
> **The baselines listed on Page 7 are insufficient.**
>
> We note that even most recent time-series SSL methods do not deal with multiple
> heterogenous datasets. The consisdered
> SSL baselines (TS2Vec, TNC, TS-TCC) in the paper which provide top performance for specific
> benchmarks where training data is used for SSL pre-training
> perform much worse than even some baselines which do not use any pre-training in our setting of pre-training on multiple datasets. This shows that even sophisticated general SSL methods cannot simply used
> to pre-train on multiple datasets.
>
> **authors provide the source code for their approach.**
>
> We have added link to the implementation code in the paper at page 7: [link](https://anonymous.4open.science/r/EmbedTS-3F5D/).
> We also plan on releasing the weights of pre-trained models on acceptance.
>
> **more results on few-shot or zero-shot learning for epidemic data..**
>
> Our work focuses on the impact of pre-training model weights for more performant training of downstream tasks. Additionally, we evaluated the efficacy of fine-tuning from smaller dataset sizes in page 8 (Q4) and Appendix Figure 4.
> Specifically we measured the performance of PEM with different fractions
> of training data and found that PEMs in on par or outperform the best baselines
> using 60-80% of the training data in benchmarks.
>
> If the reviewer is interested in any specific analysis on few-shot learning,
> we would be glad to add it to the paper.

---

### Meta-Review · Area_Chair_fx4H · 2023-12-09

**Metareview:**

The paper introduces Pre-trained Epidemic Models (PEMs) for time series forecasting of epidemic data using self-supervised learning (SSL). The reviewers provide constructive feedback on various aspects of the paper.

The strength lies in that 1) The paper is recognized for its novelty in applying SSL to pre-train models for epidemic forecasting. The significance lies in the model's ability to adapt to downstream tasks efficiently, especially in scenarios with limited data for novel diseases. 2) The paper is well-presented, and the methodology is explained clearly. The experiments are comprehensive, demonstrating the effectiveness of PEM in forecasting various diseases. 3) PEM is commended for its demonstrated efficiency in utilizing multiple training datasets while maintaining or surpassing the performance of other deep learning models. The adaptability to novel diseases is highlighted as a strong point.

The weakness lies in that 1) Reviewers express concerns about the choice of baselines, suggesting that the baselines do not represent the state-of-the-art (SOTA) performance. The absence of comparison with simpler disease-focused mechanistic models or basic auto-regressive models is noted. The paper should compare against widely used forecasting models in the epidemiological community, especially those applied during COVID-19 and influenza forecasting. 2)  Lack of information on data preprocessing steps, such as data smoothing and temporal alignment, is identified as a weakness. Detailed reporting on these steps is crucial, as they can significantly impact real-time forecasting.

Overall, the reviewers acknowledge the novelty and potential of the proposed PEMs but emphasize the need for improvements in comparison methodologies, data preprocessing reporting, and additional ablation studies, which lead to the rejection of this paper.

**Justification For Why Not Higher Score:**

The reviewers acknowledge the novelty and potential of the proposed PEMs but emphasize the need for improvements in comparison methodologies, data preprocessing reporting, and additional ablation studies, which lead to the rejection of this paper.

**Justification For Why Not Lower Score:**

N/A

---

### Decision · Program_Chairs · 2024-01-16

Reject